# Effect of Stress on Each of the Stages of the IVF Procedure: A Systematic Review

**DOI:** 10.3390/ijms25020726

**Published:** 2024-01-05

**Authors:** Anastasia Tsambika Zanettoullis, George Mastorakos, Panagiotis Vakas, Nikolaos Vlahos, Georgios Valsamakis

**Affiliations:** 1Department of Obstetrics and Gynecology, General Hospital of Rhodes, 85100 Rhodes, Greece; 22nd Department of Obstetrics and Gynaecology, Aretaieio Hospital, University of Athens, 11528 Athens, Greece; gmastorak@med.uoa.gr (G.M.); panagiotisvakas124@yahoo.gr (P.V.); nfvlahos@gmail.com (N.V.); gedvalsamakis@yahoo.com (G.V.)

**Keywords:** IVF, chronic stress, acute stress, systematic review

## Abstract

The aim of this systematic review was to examine if chronic or acute stress, measured by questionnaires or physiological biomarkers, has a separate impact on each different stage in the IVF process. A systematic search of peer-reviewed literature was performed in three databases with keywords. Preselection included 46 articles, and in all, 36 articles were included. Most studies concluded that stress has a negative effect on IVF treatment. The egg retrieval time point was most affected by chronic and acute stress. Through this research, there may be an association between chronic stress and the fertilization stage. Only chronic stress impacted the embryo transfer stage and further evidence suggested that stress decreased during this stage. The pregnancy rate stage was weakly associated with stress. Follicular cortisol was found to affect three stages. Chronic and acute stress significantly and negatively affected the egg retrieval time point. Chronic stress was associated with a lesser extent with the fertilization point, and no significant relationship between acute stress and the embryo transfer and pregnancy rate stages were found. Follicular cortisol was found to affect the process. This review contributes to the research of the relationship between stress and IVF success.

## 1. Introduction

Infertility affects millions of couples during their reproductive age. Estimates suggest that 48 million couples and 186 million individuals live with infertility globally [1], especially in developing countries.

The purpose of in vitro fertilisation is to induce ovulation in order to stimulate the ovaries to create multiple follicles before egg retrieval. These follicles are received and fertilised in the laboratory, and then as embryos transferred into the uterus. The indications for a couple to undergo an infertility treatment starts with the female tubal factor; recently, 50% of indications have been influenced by the male factor [2,3]. A successful IVF cycle has a pregnancy rate of 35%, and this is affected by maternal age, obesity, and stress [4,5,6,7]. Infertility and its treatments are a stressful situation. In particular, IVF is considered a very stressful and emotionally demanding treatment for infertile couples [2], because these couples often come from a state of repeated failed pregnancy attempts. Women with PCOS also represent a poor reproductive outcome and face distinctive difficulties and limitations in the field of ART, as well as further pregnancy complications [8,9]. Due to their problems with fertility and pregnancy, these women also experience significant psychosocial stress and reduced quality of life in comparison to those suffering from other diseases [10]. As a result, PCOS (Polycystic Ovary Syndrome) and the process of IVF (In Vitro Fertilization) can both be challenging on their own, and when combined, they may contribute to increased stress. Therefore, there is a psychological background of anxiety in relation to the general population [5,11,12].

One factor that may affect fertility is stress [13]. Stress is any emotional experience that is accompanied by biochemical, functional, and behavioral changes [14]. Its psychological onset appears with a perceived feeling of anxiety, which can be quantified with verified questionnaires, whereas the physiological changes during stress cause a state of sympathetic and hormonal changes, of which cortisol concentrations are used as a reference value [15]. In addition, stress can be categorized as either acute stress, caused by an intense condition, or chronic, where the person is permanently in a state of stress. However, this amount of stress can lead to negative consequences that may affect different organ systems [16,17], due to the “fight or flight” state of hormones [18]. However, while stress has been linked to affecting a couple’s fertility, it remains unclear whether psychological stress or functional stress, and whether acute or chronic stress, affect the effectiveness of IVF treatments. There is not enough literature to clarify whether stress is an indicator of failed treatments, and, at the same time, the methodologies for measuring stress during the IVF procedure is broad and non-identical.

It has been well documented that infertility causes stress. Additionally, there are studies and reviews which suggest that stress is associated with the assisted reproductive technology (ART) outcome. One study was conducted to evaluate levels of anxiety and depression in Chinese women undergoing in vitro fertilization (IVF) treatment [19]. The authors found that the fertilisation rate and the number of available embryos decreased as levels of anxiety increased. In another prospective study [20], the researchers discovered a potential link between stress and unfavorable in vitro fertilization (IVF) outcomes, likely influenced by psychobiological mechanism. Lastly, on the other hand, another study suggested that there was no association between psychological distress occurring before in vitro fertilization (IVF) or ICSI treatment and the occurrence of clinical pregnancy [21]. However, there are no studies showing the effect of chronic or acute stress on each time point of the IVF procedure separately, and no further studies which state which time point in the IVF procedure is affected the most. Thus, the aim of this present systematic review was to determine if chronic or acute anxiety and stress, measured by questionnaires or physiological biomarkers, have an impact on each different time point during the IVF procedure separately in women with infertility.

## 2. Materials and Methods

A thorough search of the published literature was conducted on a total of 3 databases—PubMed (MEDLINE), Science Direct, and Scopus. All retrieved articles were reviewed and their references were used to identify other relevant articles, creating a connecting source of content. These electronic databases were searched by using the following keywords: ‘IVF’; ‘acute and chronic stress’; ‘infertility female and male’; ‘oocyte retrieval’; ‘embryo transfer’; ‘IVF and acute stress and stress’; ‘egg retrieval and acute stress and chronic stress’; ‘Fertilisation and chronic and acute stress’; ‘embryo transfer and acute and chronic stress’; ‘pregnancy test and acute and chronic stress’. There were no restrictions concerning the publication year. All retrieved articles were in English and were published from 1997 to 2023. 

Concerning the selection criteria, only studies including IVF cycles as an ART method were included in the research. Animal studies were excluded and articles that emphasised minimizing stress during IVF, rather than the effect of stress, were also not considered a priority. In all, 46 research articles were selected from which 10 articles were excluded because they were not directly relevant in the scope of the present review. 

In order to establish whether articles were studying chronic and/or acute stress, the types of psychometric instruments utilized by each study were taken into consideration. Only studies that measured chronic and/or acute stress, with valid and reliable instruments, in relation to the IVF procedure, were considered. (Figure 1).

## 3. Results

### 3.1. Association of the Oocyte Retrieval Stage with Acute and Chronic Stress

A prospective cohort pilot study [23] measured stress and anxiety in 44 women undergoing IVF through a series of three questionnaires. These were the Spielberger State-Trait Anxiety Inventory (STAI), through which acute stress (STAI-S) and chronic stress (STAI-T) were measured [24]; the Perceived Stress Scale (PSS), which measures subjective distress [25]; and the Infertility Self-Efficacy Scale (ISES), which focuses on positive feelings during the infertility diagnosis and treatment [26].

These three questionnaires were completed at three time points during the course of the IVF cycle. Of the 36 women who completed a cycle, 15 participants achieved a clinical pregnancy with documentation of fetal cardiac activity at gestational age 6–7 weeks. STAI-State (acute stress) and STAI-Trait (chronic stress) scores were significantly elevated at the egg retrieval point (T2) with an even higher score amongst non-pregnant women (44.35). Women with lower scores on the STAI (34.93) and PSS (11.53), and with higher scores on ISES (106.73) questionnaire one day prior to oocyte retrieval, were more likely to achieve a pregnancy. The study concluded that all scores at T2 were a predictor of pregnancy, suggesting that acute and chronic stress can be seen as an indicating factor of the egg retrieval point [23].

In another prospective study, 151 infertile women were evaluated to determine whether baseline stress (which included acute and chronic) and procedural (acute) stress, negatively affected the pregnancy outcome [27]. This was performed through two questionnaires during the initial visit (baseline) and at the time of the procedure (procedural). To quantify these stressors, the Positive and Negative Affect Scale—PANAS [28]; the Bipolar Profile of Moods States—POMS [29]; the Perceived Stress Scale (PSS); and the Self-Rated Stress Scale, were used. 

Effects of baseline stressors on the egg retrieval point, showed that a unit increase in a woman’s chronic negative affect score on the PANAS questionnaire (e.g., a change in score from 15 to 16), was associated with a 2% decrease in the number of oocytes retrieved Moreover, there was a compounded effect of 45% decrease between the least negative group(with a score of 10) to the most negative group (with a score of 41). Thus, there was evidence that baseline stress may lead to a negative reproductive outcome. Through the analysis of acute stress measures of PANAS negative affect, and acute POMS in relevance to the oocytes retrieved, women who achieved an IVF-assisted pregnancy had similar scores with women who did not. Therefore, it was suggested that procedural stressors did not predispose women to lower chances of pregnancy or live birth delivery outcomes [27].

Another prospective study [30] was conducted with 264 women undergoing IVF between 2009 and 2010, to evaluate whether neurohormonal and psychological changes during the treatment cycle can affect the outcome. Psychological changes were assessed through questionnaires (STAI-state and C-BDI-II), whereas neurohormonal changes were evaluated through levels of norepinephrine and cortisol over four time periods. STAI-state (acute stress) scores were heightened at T2, indicating that there is a connection of stress with both pregnancy rate and live birth rate in IVF patients. Regarding the neurohormonal assessment, lower values of norepinephrine and cortisol at the time of oocyte retrieval point with values of 214.3 ng/L and 325.2 nmol/L subjectively were found in women who had successful treatments. Additionally, on the oocyte retrieval day, norepinephrine (238.3 ng/L) and cortisol (369.5 nmol/L) concentrations had statistically significantly increased *p* value < 0.05 [24], suggesting that norepinephrine and cortisol concentration at T2 might negatively affect a clinical IVF pregnancy [31].

Another study was conducted to evaluate 113 couples and investigate the psychological and hormonal changes in four time points of the IVF procedure [32]. Psychological evaluation was measured through questionnaires—Lubin’s Depression 200Adjective Check List (DACL), through which depressive mood was assessed, and the STAI [24]. There was a significant difference in anxiety levels between the non-conceiving (NC) and the conceiving (C) groups on the oocyte retrieval day. Although cortisol levels fluctuated parallel to the psychological scores during the treatment, there was not a statistically significant effect of cortisol levels of conceiving on NC women in phase II of the treatment. A significant rise in prolactin levels (PRL) at the oocyte retrieval phase was also noticeable, with a higher increase in PRL in the NC group. These findings suggested that a clear difference was demonstrated between C and NC women in terms of anxiety, yet the difference between acute and chronic stress was not specified [32].

A cross-sectional clinical study studied 49 women in order to assess the association between sensitivity to stress and IVF [33]. On the day of the egg retrieval, all women took part in a Stroop color and word test. Systolic and diastolic blood pressure and heart rate measurements were taken at baseline, during the time of the test, and 10 min after the end of testing. Findings showed that an elevated systolic blood pressure (SBP) and heart rate (HR) were present at the egg retrieval point. Moreover, patients with a successful outcome had a lower percent increase in both SBP and HR than those in the unsuccessful group. These findings may suggest that cardiovascular stress is congruent with a poor treatment outcome [33].

Conversely, some studies do not confirm the relationship between anxiety and the treatment outcome of IVF. A prospective cohort study examined 72 patients undergoing IVF in 2017 and 2018 [34]. Physiological stress was assessed by salivary cortisol before oocyte retrieval, as well as through follicular cortisol. Emotional stress was then quantified with the STAI questionnaire and a 1–10 Visual Analogue Scale (VAS), which is used to assess perceived stress [34]. This study showed that salivary cortisol concentrations increased by 28% from pretreatment phase (0.46 ± 0.28 μg/dL) to a maximum concentration on the oocyte retrieval day (0.59 ± 0.29 μg/dL), with a parallel significant increase of 39% in stress scale scores. On oocyte retrieval day, cortisol concentrations were above the general population average concentration (0.5 μg/dL). Yet, this acute increase had no effect on the reproductive outcome. A significant positive correlation was found between follicular cortisol concentrations on oocyte retrieval day and fertilisation rate. 

Therefore, this study concluded that acute stress at the oocyte retrieval day did not negatively affect IVF outcomes [34]. Moreover, it has been assumed that high follicular fluid cortisol concentrations during the egg retrieval phase might have positive effects on fertilisation rate [35].

A meta-analysis was used to investigate a total of 11 studies to determine whether anxiety and depression scores during assisted reproductive technology treatment (ART) are associated with treatment outcomes, as well as if changes between baseline measurements and during treatment further affected the outcome [36]. Depression and state anxiety, assessed during various time points in of the ART treatment cycle, were able to predict the ART outcome, particularly the oocyte retrieval stage. However, there was no evidence that the change in levels of depression or state anxiety from baseline to during treatment (T2) was correlated with the ART outcome. Additionally, it has been observed that the number of studies examining baseline measures versus during ART treatment was relatively small [37].

### 3.2. Association of the Fertilisation Stage with Acute and Chronic Stress

A study investigated 151 women to determine whether baseline and procedural stress affect the rate of fertilisation, measured as the number of oocytes fertilised [27]. All women completed two questionnaires which determine the stress level where higher scores indicate a higher level of stress and anxiety [27], during the first clinic visit and immediately prior to embryo transfer. The study showed that changes in scores of baseline stress at the initial clinic visit affected the number of oocytes fertilized [27]. More specifically, each unit increase in women’s scores on the PANAS and POMS scales had a negative predictive value for the number of oocytes fertilised. On the contrary, the higher the women ranked on their likelihood for success, the greater the numbers of oocytes fertilised. 

When assessing acute stress, it was established that each score increased by one (1) in the acute PANAS scale led to a decline by 2% of the number of oocytes fertilised. This study further determined that although procedural stressors did affect the fertilised oocytes, women who achieved pregnancy had similar PANAS, POMS scores with women who had had a failed IVF treatment. This study concluded that the level of procedural stressors observed during IVF did not affect the chance of pregnancy or live birth delivery, and that baseline stress is the most critical time for monitoring and decreasing stress levels, and may affect the treatment outcome [27]. 

A prospective cohort studywas designed, in which 48 women participated [38]. This study evaluated the influence of psychological factors on the outcome of the fertilisation step in IVF treatment. Each woman was assessed psychologically on the day before oocyte retrieval (OR) with the Child Project Questionnaire (CPQ) and with the Ways of Coping Checklist, which deals with the internal and/or external demands of specific stressful encounters [39]. In addition, the State-Trait Anxiety Questionnaire was completed by all women two days prior to OR, while the State Anxiety Questionnaire form was completed again for 6 more days. When analyzing the child project questionnaire, the score on women’s factor II, evaluating the perception of marital harmony with the prospect of conceiving a child, was significantly higher in women with at least one embryo than in women with an unsuccessful fertilisation [39].

Stress measured through the STAI-state inventory did not affect the fertilisation rate, yet after women were informed about the failure of fertilisation, the anxiety levels elevated significantly. Trait anxiety scores were also not significantly different between the successful and unsuccessful groups, suggesting that stressors during the procedure may not affect the fertilisation step. Chronic stress factors were not acknowledged as the measurements of stress commenced at a procedural rather than a baseline, or chronic, time point [34]. 

A recent prospective cohort study of 72 patients undergoing IVF in 2017 and 2018 evaluated physiological stress by salivary cortisol measurements before oocyte retrieval and emotional stress, with the State-Trait Anxiety questionnaire and a 1–10 Visual Analogue Scale [35]. The researchers found higher concentrations of cortisol measurements and stress level reports on oocyte retrieval day, but with no effect on the fertilisation rate, indicating that women’s anxiety and stress level may not affect their fertilisation rate and their IVF outcome [35]. This study also found that higher follicular fluid cortisol was related to higher fertilization rate and proposed that further research in this area is warranted.

### 3.3. Association of Embryo Transfer with Acute and Chronic Stress

A prospective study in respect to the Embryo Transfer point results showed that, a 1-point increase in POMS depression and hostility scales was negatively predictive in the number of embryos transferred [27]. Women with a score of 12 for example, meaning they were the most depressed and hostile, had approximately one to two fewer embryos transferred compared to the least depressed women with a score of 3. An increase of 16 points in the POMS total scale, which measures the total mood disturbance of mood states [40,41], was associated with one less embryo transfer. Additionally, a 75% increase in one’s optimism about their pregnancy resulted in one extra embryo being transferred. When measuring chronic stress, this study determined that at baseline, the number of embryos transferred decreased, along with every increase in women’s negative affect scores on the PANAS and POMS scales [27].

Furthermore, in this study [27], there was a significant increase in perceived stress, or feelings or thoughts of the individual about how much they feel at a given point or period of time [42]. Therefore, positive affect decreased both after hormone treatment at the initial clinic visit and during the time of the procedure. According to this study, hormones used in IVF do play a role in mediating emotional responses and that baseline (chronic and acute) stress may affect biologic end points [27].

A study investigated the psychological and hormonal changes during different time point of the IVF treatment, in order to examine any relationship between the psychological variables of anxiety and depression and the levels of PRL and cortisol, during the hormonal stimulation, oocyte retrieval, embryo transfer, and pregnancy confirmation blood test phases [32]. The third measurement point took place immediately prior to the embryo transfer. Neither group showed differences in scores between levels, nor this finding indicated that the time point of the procedure was not affected by any stressors. However, this study proposed that this may be due to the fact that the previous time points of the procedure were successful and thus, diminished or eliminated anxiety [32].

A prospective study was conducted to investigate the pregnancy outcome of women undergoing assisted reproductive technology (ART), evaluated through a combination of psychological stress examinations on the day of embryo/blastocyst transfer [36]. In total, 114 women underwent in vitro fertilisation (IVF) from April 2012 to May 2012 and were examined before the transfer by gathering salivary secretion. A-amylase and cortisol concentrations were then quantified by using biochemical methods. In addition, participants were asked to answer a General Health Questionnaire (GHQ28) and Zung’s Self Rating Depression Scale (SDS) after the transfer. Results were then compared between the NC and C groups. Findings showed that the GHQ28 and SDS scores were parallel between the two groups, while the same happened for the salivary concentrations. To conclude, Purewal et al. [36] did not find any significant differences in the molecular stress marker scores or in the psychometric instruments between the pregnant and non-pregnant groups at the embryo transfer point. 

A prospective cohort study of 72 patients undergoing IVF in the period of 2017–2018 assessed physiological stress by salivary cortisol measurements during important points during the IVF process, including the embryo transfer [35]. Emotional stress was analyzed with the STAI Inventory and with a 1–10 VAS. Any correlations between cortisol concentrations, psychological stress and IVF outcome were evaluated. Salivary cortisol concentrations decreased by 29% on embryo transfer day. The Stress Scale score increased by 39% between pretreatment and oocyte retrieval day, and then decreased by 12% on embryo transfer day, suggesting that acute stress is present during the process, yet fluctuates during different time points. Salivary cortisol and Stress Scale were not related to the embryo transfer, and it can be concluded that physiological and psychological stress may not negatively affect this time point [35]. 

### 3.4. Association of Pregnancy Test Day/Rate with Acute and Chronic Stress

A prospective study between 1993 and 1998 to evaluate whether baseline or procedural stress, during in vitro fertilisation (IVF) affects pregnancy or live birth delivery rates. Results showed that when examining baseline stressors, stress measures had negative effects on the outcomes of successful pregnancy and live birth delivery [27]. More specifically, a unit increase in acute positive affect on the PANAS scale was associated with a 7% lower risk of no live birth delivery. Therefore, the risk of no live birth delivery was 93% lower for women who scored highest on the positive affect scale, versus those who scored lowest. An analysis of the effects of procedural stress (mood, affect) on pregnancy and live birth delivery, showed that there were no statistically significant associations between any of the stress measures at the time of the procedure, on the outcomes of successful pregnancy and live birth delivery. This study concluded that baseline stress affected pregnancy and live birth delivery, whereas procedural stress only influenced biologic end points [27]. 

A prospective study was performed between 2009 and 2010 to assess the relationship between psychological stress and reproductive outcome in 264 women undergoing IVF [30]. The researchers aimed to evaluate whether psychological stress, as well as changes in hypothalamus-pituitary-adrenal (HPA) axis and sympathetic nervous system (SNS), at different time points during a first IVF cycle, correlated with the reproductive outcome. Through psychological questionnaires state anxiety was measured at the day of pregnancy detection (T3) as well as through neurohormonal serum measurements of norepinephrine and cortisol. At the pregnancy detection day (T3), the non-pregnant women reported higher anxiety and depression scores compared to the pregnant group. Concentrations of NE and cortisol in serum at T3 were higher in the non-pregnant group, and NE levels at the day of pregnancy detection and cortisol concentrations at the early pregnancy had a significant correlation with State Anxiety scores [30]. This stress-induced hormone deficiency may cause maternal biological responses, including localized inflammation in uterine tissue and sustained depression of progesterone production, challenge the endocrine–immune steady state during pregnancy, leading to serious consequences for the fetal environment significantly affecting the pregnancy outcome [30].

A statistically significant association between state anxiety and live birth rate in IVF-pregnancies was found, which adjusted for biological factors–female age, BMI, previous miscarriage, endometrial thickness, and number of transferred embryos. In conclusion, anxiety was associated with both pregnancy rate and live birth rate in IVF patients, an effect that was partly conducted by processes in the HPA and SNS and this finding suggests that norepinephrine and cortisol concentrations may negatively affect the clinical pregnancy rate of IVF treatment [30].

A prospective cohort study of 72 patients undergoing IVF was designed [29]. Physiological stress was assessed by salivary cortisol measurements at three time points where follicular cortisol was also assessed. Emotional stress was evaluated at each measurement with the STAI Inventory and with a 1–10 VAS scale. Correlations between cortisol concentrations, psychological stress and IVF outcome were assessed. Salivary cortisol and Stress Scale at the pretreatment phase were not related to clinical pregnancy rate. The research cautiously concluded that physiological and psychological stress did not negatively affect IVF outcomes; however, high follicular cortisol concentrations may have positive effects on pregnancy rates [35].

A systematic review and meta-analysis [43], aimed to analyze anxiety and depression among infertile women at different time points during the first (IVF) or intracytoplasmic sperm injection (ICSI) treatments. The researchers compared the measurement outcomes at three time points: before treatment (T0), after pregnancy detection (T2), and one to six months after treatment (T3). A comparison of the 400different time points between the pregnant women and the non-pregnant women, showed that stress at T2 was significantly higher compared to T0. At T2, non-pregnant women also reported higher levels of anxiety compared with pregnant women. This led the authors [37] to conclude that anxiety in infertile women undergoing their first IVF, were associated with the T2 stage; yet this study did not address whether chronic or acute anxiety affected pregnancy detection.

A nested case–control study [21] aimed to investigate the association of pre-treatment psychological distress and clinical pregnancy rates among infertile couples undergoing IVF or ICSI treatments in November 2015 and January 2019. A total of 150 women with no clinical pregnancy after their first IVF or ICSI embryo transfer were identified as cases, and a total of 300 women the same age, who had a clinical pregnancy were the controls. Results showed that there was no statistically significant association between maternal psychological stress scores and symptoms and clinical pregnancy. These results led to the conclusion that pre-treatment anxiety and chronic anxiety did not have a clinical effect on pregnancy rate [21].

Finally, another prospective study aimed to examine the effects of stress and depression symptoms before IVF treatment on clinical pregnancy rates [44]. The sample included 142 women undergoing IVF treatment at Royan Infertility Clinic in Iran. Data was collected between February and March 2017. All women completed the Hospital Anxiety and Depression Scale (HADS), measuring symptoms of anxiety and of depression. The PSS-10 was also used, that measures overloading situations over the past month. In this study, the clinical pregnancy rate was 26.8%. There were no significant differences between pregnant and non-pregnant women with respect to anxiety, depression, and stress scores, and this led to the conclusion that there was no relationship between the IVF clinical pregnancy rate and anxiety, and depression symptoms. Both simple and multiple analyses indicated that stress symptoms had no significant effect on the IVF outcome [44].

## 4. Discussion

This systematic review examined effects of chronic or acute anxiety and stress on the IVF cycle, specifically during different time points of IVF. Stress was assessed psychologically through psychometric instruments; biologically via stress molecules such ascortisol, prolactin and norepinephrine levels; and clinically through cardiovascular parameters, such as systolic and diastolic blood pressure and heart rate. It has long been assumed that infertility is stress related. While a number of studies have looked at stress over the course of the IVF cycle, the present review aimed to analyse evidence that associates chronic and acute stress with each IVF time point separately. Most of the studies in this review concluded that stress and anxiety negatively affected IVF treatment. 

According to the studies, it seems that the egg retrieval point is the stage that is mostly affected by both chronic and acute stress during the procedure. Indeed, studies measuring molecules and using psychological scales showed both chronic and acute stress effect the egg retrieval point. Specifically, higher scores of anxiety and stress led to a negative outcome in the number of oocytes retrieved [16]. Additionally, women with a negative pregnancy outcome had higher levels of stress biomarkers such as cortisol at an earlier egg retrieval time point [30].

Lastly, patients who demonstrated a lower percent increase in both SBP and HR at the egg retrieval point had successful outcomes, suggesting that cardiovascular stress is associated with poor treatment outcomes. This evident association between anxiety and the egg retrieval point might be explained by the fact that this is the first stage of the IVF cycle. Thus, women are stressed due to their infertility as a couple, as well as the outcome of the procedure. Additionally, as an experience, the egg retrieval point is a minimally invasive procedure, where the oocytes are extracted through the follicular tube. Therefore, the nature of the procedure may create additional stress. In this time point women with PCOS are accompanied with the risk of ovarian hyperstimulation syndrome (OHSS) [45]. As a result, this difficulty in this time point particularly for women with PCOS syndrome can contribute to heightened stress levels.

The next time point examined was the fertilisation point. Chronic stress was found to be associated with the fertilisation point in three studies. Through this research, it is concluded that there may be an association between chronic stress and the fertilisation process. One study showed that changes in baseline chronic stress scores affected the number of oocytes fertilized [27]. Other studies assessing the effect of stress during the fertilisation process did not specifically examine this time point as an outcome; therefore, not enough research could be found that could corroborate or disprove the relevant findings. In this sense, further research is needed that examines the effect of stress during fertilisation. This time point may not be associated with stress, due to the fact that it is an in vitro procedure which includes no patient interaction. Yet, it is a time point where further studies could be performed, given that stress might affect the quality of the oocytes, thus affecting the fertilisation outcome. Although one study suggested that both chronic and acute stress might affect the fertilisation outcome [27], all other relevant studies published to date suggested that there were no statistically significant differences [46].

Interestingly, some research suggested that not only did stressors not affect the embryo transfer as a time point, but that the level of anxiety actually decreased [32]. This may be explained given that this is the end time point of the procedure, and therefore women may have grown more accustomed to the process. Additionally, women during this stage have been informed that egg retrieval and fertilisation have been successful, and this knowledge likely creates a positive affective response during this first part of the treatment. 

The last time point reviewed was the pregnancy rate, specifically a positive pregnancy result, or a success in clinical pregnancy and live birth rate, and its association with acute or chronic stress. Only one study associated acute stress with the pregnancy rate [30]. In another study, baseline stress was associated with pregnancy rate, but acute stress had no significant difference between pregnant and non-pregnant women [27]. All remaining studies showed no statistically significant differences. 

The live birth rate was found to be associated with both chronic and acute stress; however, it was unclear whether stress during or before the IVF process is related tothe live birth rate, because other factors are also involved during pregnancy. Specifically, the operationalization of a diverse array of psychometric instruments by a number of studies made it difficult to ascertain whether they were measuring chronic or acute stress. This led the researchers of this review to further research the available literature, but this practice did not produce studies that fit the search criteria, especially concerning the different time points of IVF. Some of the research in this systematic review suggested that follicular cortisol may have an impact in egg retrieval, fertilization, and pregnancy time points. Studies indicated that higher follicular fluid cortisol concentrations were indeed related to higher rates of fertilisation, egg retrieval rate, and pregnancy. This review also revealed no correlation regarding salivary and follicular cortisol concentrations. It is proposed that this research area should be expanded, because even though higher follicular fluid is a promising marker, the studies reviewed were not explicit nor based on follicular cortisol. Follicular fluid can be a predictor of success of the IVF treatment, therefore creating a prognostic biomarker. This is an issue that needs to be further analyzed in prospective studies as a predictive biomarker of stress in IVF success in every timepoint separately. 

Regarding women with PCOS, they have a decreased reproductive outcome in comparison to the women studied in this review. More specifically, regardless the unovulatory cycles induced by these women, there is a discussion that the presence of co-morbidities is associated with changes in the quality of both oocytes and endometrial tissue, as well as disruptions in the communication between the endometrium and embryos. This alteration raises the likelihood of infertility and heightens the risks of complications in both early and late stages of pregnancy through an abnormal trophoblast invasion and placentation [8]. One study concluded that the characteristic features of PCOS, including hyperandrogenism, obesity, insulin resistance, and metabolic abnormalities, may collectively contribute to the elevated risk of obstetric and neonatal complications such as pregnancy-induced hypertension and pre-eclampsia and a higher likelihood of premature delivery [9]. On the other hand, another study discovered that PCOS is associated with significant psychosocial stress and reduced quality of life in comparison to other diseases [10]. The challenges of managing PCOS symptoms, concerns about fertility, and the impact on emotional well-being can contribute to stress. Yet, despite this special interest in this population, we did not find studies on the effect of acute or chronic stress in each of the stages of the IVF procedure in women with PCOS. This is lacking in the bibliography, as it would be of interest to discuss the PCOS population amongst infertile women. Therefore, future studies should address this issue.

More research is warranted so that the effects of chronic and acute stress during different time points of the IVF procedure can be better understood, and stress relieving actions and interventions can be pursued at the correct moment. Certainly, when discussing the potential beneficial effects of supplements, particularly inositol and other compounds with reported effects on brain health, it is valuable to provide the potential benefits these antioxants might have in reducing stress. More specifically, Inositol is a compound that has been studied for its potential role in improving various aspects of reproductive health, and there is some evidence suggesting that it may have benefits in the context of both fertility and stress reduction. One study suggested that inositol may be beneficial for depressed patients, especially those with PMDD [47]. It has also been suggested that the therapeutic activity of inositol is involved in the regulation of neurotransmitters in the brain, including serotonin, a neurotransmitter that plays a role in mood regulation, and imbalances have been associated with conditions such as anxiety and depression [47]. This could be helpful in PCOS patients who are prone to anxiety and depression.

On the other hand, another study measured the serum level of fatty acid on days 3–9 of stimulation and concluded that high levels of total omega-3 PUFAs and EPA are associated with higher probability of pregnancy and live birth [48]. Moreover, Omega-3 fatty acids, found in fish oil and flaxseed oil, have anti-inflammatory properties, and may support cognitive function. More specifically, some studies suggest that omega-3 fatty acids, including DHA, may help modulate cortisol levels in response to stress thus promoting mental health [48]. In particular, it was established that the relationship between HPA-axis and fatty acids showed significant differences in recurrent MDD patients versus controls [49].

Other supplements such as Vitamin D have also been associated with mood regulation. Specifically, a clinical trial was conducted with 158 females (aged 15–21 years) with PMS-related severe symptoms of the emotional and cognitive domains and serum 25(OH)D levels [50]. Vitamin D or placebo was prescribed for 4months, and this study concluded that Vitamin D therapy was effective for improving anxiety. A metanalysis concluded that sufficient vitamin D status is associated with better outcomes in IVF [51]. More specifically they found that women with sufficient vitamin D had significantly higher biochemical pregnancy, ongoing pregnancy, and live birth rates [51].

B vitamins, including B6, B12, and folic acid, play a role in neurotransmitter synthesis and may influence mood. A study was conducted in 478 young adults with self-reported anxiety [52]. This study found that adding Vitamin B6 through supplementation resulted in a decrease in self-reported anxiety and showed a tendency toward reducing depression. Moreover, it led to an increase in the surround suppression of visual contrast detection. However, its impact on other measured outcomes was not consistently significant. On the other hand, Vitamin B12 supplementation exhibited trends suggestive of alterations in anxiety levels and visual processing [52]. Ensuring adequate levels of these vitamins may contribute to overall well-being and stress management while simultaneously affecting the prenatal period [52].

Magnesium is also involved in the regulation of the body’s stress response. Magnesium is involved in the synthesis and function of neurotransmitters, including serotonin and gamma-aminobutyric acid (GABA), which play key roles in mood regulation. Deficiency in magnesium may affect neurotransmitter balance and contribute to symptoms of stress and anxiety [53]. A study, reported positive effects of 12 weeks intake of 75 mg Mg combined with Hawthorn (75 mg) and California poppy (20 mg) extracts vs. a placebo in individuals reporting mild anxiety or symptoms of general anxiety disorder [54].

Lastly, melatonin is a hormone that is naturally produced by the pineal gland in response to darkness, helping regulate the sleep–wake cycle [55]. Some research has explored the potential role of melatonin supplementation in the context of in vitro fertilization (IVF). One study found that the mean number of the retrieved oocytes and the embryo ratio were significantly higher in the melatonin-administered group (group A) than that the non-administered group (group B), concluding that for individuals undergoing the stress of IVF treatments, melatonin supplementation might contribute to stress reduction but the sleeping problem itself may not be fixed [55].Finally, further study is proposed in order to clarify the extent to which follicular cortisol has an effect on the IVF process.

In addition to the previous described studies assessing supplements that could ease stress during the IVF procedure in infertile women, it would be of interest to assess studies dealing with the PCOS population specifically. It is essential to consider the influence of factors such as endometriosis and obesity, which can contribute to subfertility or infertility, as well as factors like nulliparity and advanced age, which are indicators of a less favorable prognosis. This consideration is crucial for customizing an optimal treatment approach in the context of fertility [56]. Dealing with infertility can also be emotionally challenging, and women with PCOS may experience heightened psychological stress. The uncertainty of achieving a successful pregnancy, coupled with the physical and hormonal challenges of PCOS, can contribute to elevated stress levels. Understanding the specific challenges, they face in terms of reproductive outcomes and pregnancy complications is crucial for effective management and support. One study [57] suggested that various supplements, such as myo-inositol, can augment the positive effects of lifestyle modifications and potentially improve the outcomes of in vitro fertilization (IVF). Specifically, they may enhance factors such as oocyte yield and the likelihood of achieving a successful pregnancy. Regarding stimulation protocols for IVF, utilizing antagonist cycles along with carefully administered GnRH agonist triggers, pre-treatment involving metformin, and ensuring adequate levels of vitamin D can be effective strategies to mitigate the associated risks of ovarian hyperstimulation syndrome (OHSS) [45]. Moreover, these supplements such as myo-inositol and vit D have been suggested to potentially contribute to stress reduction, particularly in the context of reproductive health, as mentioned previously. As a result, in these patients the treatment may reduce psychological stress, which is already very high and ameliorate the treatment outcome.

The present review concludes that stress and anxiety largely affect the egg retrieval point, at a chronic and acute level, and that failure at this time point is positively associated with high anxiety scores and biomarkers. Chronic stress was primarily associated with fertilisation and the embryo transfer points and with the pregnancy rate. The results from this study contribute to the results in a relatively ambiguous area of research on the relationship between stress and IVF success. In consideration of the stress induced by fertility-enhancing procedures, the progression of intensity and complexity of treatment is, probably, the cornerstone of the management of PCOS-related and non PCOS-related infertility [56]. In order to reduce stress associated with infertility and IVF, larger, well-designed studies in humans are further warranted to incorporate different stress relieving compounds into the clinical management of Stress during the IVF procedure.

## Figures and Tables

**Figure 1 ijms-25-00726-f001:**
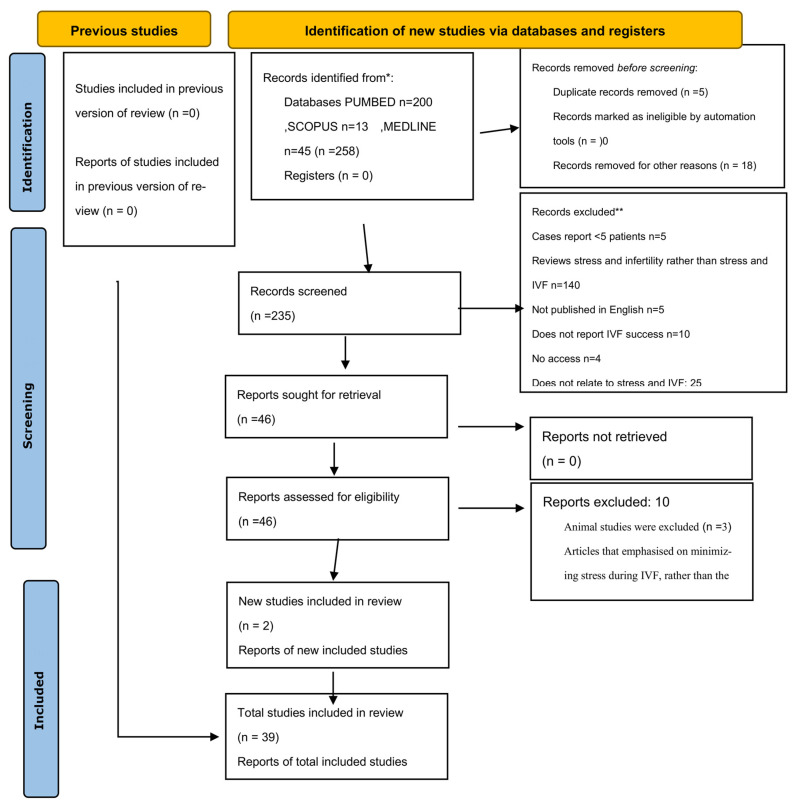
Prisma Flowchart. * Consider, if feasible to do so, reporting the number of records identified from each database or register searched (rather than the total number across all databases/registers). ** If automation tools were used, indicate how many records were excluded by a human and how many were excluded by automation tools. From: Page et al., 2021 [22].

## Data Availability

The data that support the findings of this study are openly available in data bases where all articles were retrieved, such as PubMed (MEDLINE), Science Direct, and Scopus.

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
