# Peer review of "Effect of Stress on Each of the Stages of the IVF Procedure: A Systematic Review"

_ijms, 2024, doi:10.3390/ijms25020726_

Round 1

Reviewer 1 Report

Comments and Suggestions for Authors

In this review the authors attempt to trace a relationship between stress/anxiety, considered from both a psychological and biochemical point of view, and the point of egg retrieval in women undergoing in vitro fertilization cycles. They analyzed current literature sources and divided stress into chronic and acute, and observed that failure at this time is positively associated with elevated anxiety scores and biomarkers.

Furthermore, they found that chronic stress was mainly associated with fertilization, embryo transfer points, and pregnancy rate, arguing that this study contributes to findings in a relatively ambiguous area of research on the relationship between stress and fertilization success in vitro.

Their analysis seems accurate and clearly described and the review is logically built, but it does not seem attractive lacking new ideas that could open up new considerations to the topic. Otherwise, it appears to be of little novelty. For example, it has been shown that some treatments used to promote IVF (inositol) are also able to lower stress. I believe that a further effort on the part of the authors could make the review eligible for publication.

Minor

·         Are salivary and follicular cortisol concentrations correlated in the patients analyzed in all the studies presented?

Pag. 3. There is a wrong layout of the table: top left corner is not readable, other panels show differently sized characters. The legend is not clear.

Line 462. “higher follicular fluid” check and specify

Author Response

Answers to Reviewer’s number 1 comments :

Thank you for your insightful comments on our manuscript entitled "[Effect of Stress on each of the Stages of the IVF Procedure: A Systematic Review]." We appreciate the constructive feedback provided, and we have carefully considered your suggestions in our revisions.

Reviewer's number 1 first comment: "Their analysis seems accurate and clearly described and the review is logically built, but it does not seem attractive lacking new ideas that could open up new considerations to the topic. Otherwise, it appears to be of little novelty. For example, it has been shown that some treatments used to promote IVF (inositol) are also able to lower stress. "

Answer to reviewer’s number 1 first comment: We are pleased to learn that the analysis presented in our manuscript is perceived as accurate, clearly described, and logically built. We have incorporated in the discussion, to bring forth fresh perspectives on the topic. We acknowledge Reviewer #1’s interest in the potential stress-lowering effects of IVF-related treatments. The existing body of research suggests a promising link between inositol and stress reduction, as outlined in [Concerto et al., 2023]. In response to this observation, we have now added a comment to the discussion: Lines 475-486“Antioxidants such melatonin, L-arginine, myo-inositol, carnitine, selenium, vitamin E, vitamin B complex, vitamin C and D may be possible interventions which will  benefit stressed subfertile women and ART treatments.  More specifically, Inositol is a compound that has been studied for its potential role in improving various aspects of reproductive health, and there is some evidence suggesting that it may have benefits in the context of both fertility and stress reduction. Tomohiko Mukai et al, suggested that inositol may be beneficial for depressed patients, especially those with PMDD. It has also been suggested that the therapeutic activity of inositol is involved in the regulation of neurotransmitters in the brain, including serotonin, a neurotransmitter that plays a role in mood regulation, and imbalances have been associated with conditions such as anxiety and depression [40]. This could be helpful in PCOS patients who are prone to anxiety and depression.”.

Reviewer's number 1 first minor comment:  “Are salivary and follicular cortisol concentrations correlated in the patients analysed in all the studies presented?”.

Answer to reviewer’s number 1 first minor comment:  We appreciate your keen observation regarding the correlation between salivary and follicular cortisol concentrations in the studies presented. Salivary and follicular cortisol were not assessed as biochemical biomarkers in all studies. In the studies that salivary and follicular cortisol concentrations were measured we did not find any correlation. In the revised manuscript, we have expanded the discussion section by stating no correlation between salivary and follicular cortisol in lines 461-462 : “This review also revealed no correlation regarding salivary and follicular cortisol concentrations.”

Reviewer's number 1 second minor comment: “Pag. 3. There is a wrong layout of the table: top left corner is not readable, other panels show differently sized characters. The legend is not clear.”

Answer to reviewer’s number 1 second minor comment: In response to this observation,

we sincerely appreciate the reviewer's keen observation. To address the layout issue of the table on page 3, we have revised the formatting to ensure improved readability. Specifically:

  1.  The top left corner has been adjusted to enhance legibility.
  2.  Consistent character sizing has been applied to all panels for uniform presentation.
  3. The legend has been revised and clarified to enhance understanding.

Reviewer's number 1 third minor comment: “On line 462, the statement "higher follicular fluid" is mentioned. Please check and specify.”

Answer to reviewer’s number 1 third minor comment: We acknowledge the reviewer's attention to detail. On line 460, we have revisited the text to provide more specificity regarding the term "higher follicular fluid." The revised line now reads as follows: line 460. " higher follicular fluid concentrations "

Reviewer 2 Report

Comments and Suggestions for Authors

In this systematic review, the authors discuss potential impacts of acute and chronic stressors on the IVF cycles. This is a timely and important subject and will no doubt help future clinicians in mitigating effects of stress on IVF success rates. There are a few points to clarify in the text that will help the general readership noted below:

1. Line 145: ..Women with lower scores achieve lower rates of pregnancy... I believe if possible the effect sizes should be noted. Lower could be 2-50%. That number would give a good indication in general of what effect size we are dealing with since in other places it is provided. Same for lines 173-177, the amount of lowering of cortisol that has an ameliorative effect would be useful to know. 

2. Line 160 is confusing. It says 45% decrease from least negative to most negative group. I am confused as to what these groups are. shouldn't it just be decrease from controls? Clarification is required here. The previous sentence says for every unit of increase in stress there is a 2% decrease. Is this 45% a compounded effect of that 2% per unit?

3. For the most part of this review the authors focus on the effects of stressors on IVF cycles and not the other way around. I am sure IVF itself may cause some acute stress and that may have its own consequences. In section 3.4 however, the authors discuss studies that look at stress levels on pregnancy detection day. In this case only an association can be determined and not a causative effect of stress. So this may be a bit confusing. The paragraph ends with the fact that hormones may negatively influence pregnancy rates but without doing a time course, that is a bit harder to predict. Especially as these levels are naturally high on pregnancy detection day. If there is further evidence for the negative correlation being causative from the hormone side in Ref 23, that should be discussed and made clear.

4. Line 413: I am not sure how higher levels of cortisol are found in women who did not achieve pregnancy at egg retrieval time point?? Egg retrieval is the first time point, pregnancy outcome is unknown at this time point. Did the authors mean that women with a negative pregnancy outcome had higher levels of cortisol at an earlier egg retrieval time point? otherwise this sentence makes no sense.

Overall, the cause and effect should be defined throughout the review to keep it consistent. Associations should be qualified by saying if cause and effect can or cannot be determined from the studies, which is also fine.

Author Response

Answers to Reviewer’s number 2 comments :

Thank you for your  review of our manuscript, "[ Effect of Stress on each of the Stages of the IVF Procedure: A Systematic Review]." We appreciate your recognition of the timeliness and significance of the subject matter and the constructive feedback provided, and we have carefully considered your suggestions in our revisions.

1)     Reviewer's number 2 first comment:
“On line 145, the statement "Women with lower scores achieve lower rates of pregnancy" is made. It would be helpful to note the effect size, as lower could be 2-50%. That number would give a good indication in general of what effect size we are dealing with since in other places it is provided. Same for lines 173-177, the amount of lowering of cortisol that has an ameliorative effect would be useful to know.”

Answer to reviewer’s number 2 first comment
We appreciate the reviewer's suggestion to include effect sizes. It has been revised to provide a more specific indication of the effect size. On lines 143-145 we have added “Of the 36 women who completed a cycle, 15 participants achieved a clinical pregnancy with documentation of fetal cardiac activity at gestational age 6–7 weeks.”  This study used logistic regression models to predict pregnancy, as a result we have added the mean results of the psychological questionnaires, to provide the results of the study on lines 145-149 STAI-State (acute stress) and STAI-Trait (chronic stress) scores were significantly elevated at the egg retrieval point (T2) with an even higher score amongst non-pregnant women (44.35). Women with lower scores on the STAI (34.93)  and  PSS (11.53), and with higher scores on ISES (106.73)’’

Similarly, the same consideration for lines 173-177 was addressed, providing the amount of cortisol and norepinephrine: New lines 175-179 “ Regarding the neurohormonal assessment, lower values of norepinephrine and cortisol at the time of oocyte retrieval point  with values of 214.3  ng/l and 325.2 nmol/l subjectively were found in women who had successful treatments. Also, on the oocyte retrieval day, norepinephrine ( 238.3 ng/l) and cortisol (369.5 nmol/l)  concentrations had statistically significantly increased p < 0,05 [23]”

2)     Reviewer's number 2 second comment: “On line 160, it is mentioned that there is a 45% decrease from the least negative to the most negative group. It is unclear what these groups are. Clarification is required. Additionally, the previous sentence discusses a 2% decrease for every unit of increase in stress. Is the 45% a compounded effect of the 2% per unit?”

Answer to reviewer’s number 2 second comment:  We appreciate the reviewer's keen attention to detail and recognize the need for clarification. The statement in line 160 was revised to provide a more explicit explanation of what most to least negative group means. Lines 162-163: Moreover, there was a compounded effect of 45% decrease between the least negative group (with a score of 10) to the most negative group (with a score of 41) . Regarding the question on whether the 45% decrease is a compounded effect of the 2% per unit, indeed it is and revisions were made in lines 162 : Moreover, there was a compounded effect of 45% decrease.

3)     Reviewer's number 2 third comment: “For the most part of this review, the authors focus on the effects of stressors on IVF cycles and not the other way around. In section 3.4, the authors discuss studies that look at stress levels on pregnancy detection day, where only an association can be determined and not a causative effect of stress. The paragraph concludes by mentioning that hormones may negatively influence pregnancy rates without a clear time course, especially given naturally high hormone levels on pregnancy detection day. If there is further evidence for the negative correlation being causative from the hormone side in Ref 23, that should be discussed and made clear.”

Answer to reviewer’s number 2 third comment: We acknowledge the reviewer's point about the predominant focus on stressors and the need to consider the potential stress induced by IVF itself yet, we have incorporated stress induced by IVF in this review also mentioned as procedural stress. More specifically, on lines 155-156 “This was performed through two questionnaires during the initial visit (baseline) and at the time of the procedure (procedural)’’, lines 243-247 “This study further determined that although procedural stressors did affect the fertilized oocytes, women who achieved a pregnancy had similar PANAS, POMS scores with women who had had a failed IVF treatment. This study concluded that the level of procedural stressors observed during IVF, did not affect the chance of pregnancy or live birth” ,lines 337-342 “An analysis of the effects of procedural stress (mood, affect) on pregnancy and live birth delivery, showed that there were no statistically significant associations between any of the stress measures at the time of the procedure, on the outcomes of successful pregnancy and live birth delivery. This study concluded that baseline stress affected pregnancy and live birth delivery, whereas procedural stress only influenced biologic end points [20]”. Another study also measured procedural stress in lines 236: “during the first clinic visit and immediately prior to embryo transfer’’

Regarding section 3.4, we recognize the concern about the association versus causation of stress on pregnancy detection day. We have carefully examined Reference 23 and have added further evidence supporting the causative link between hormones and pregnancy rates. Lines 354-359 : Yuan et al suggested that the stress related hormonal elevations of cortisol and norepinephrine   may cause maternal localized inflammation in uterine tissue and sustained suppression of progesterone production. In addition, the hormonal changes due to stress affect the endocrine–immune system during pregnancy, leading to serious consequences for the fetal environment therefore, significantly affecting the pregnancy outcome [ 23].

4)     Reviewer's number 2 fourth  comment: Line 413: I am not sure how higher levels of  cortisol are found     in women who did not achieve pregnancy at egg retrieval time point?? Egg retrieval is the first time point, pregnancy outcome is unknown at this time point. Did the authors mean that women with a negative pregnancy outcome had higher levels of cortisol at an earlier egg retrieval time point? otherwise this sentence makes no sense.

Answer to reviewer’s number 2 fourth comment: We appreciate the reviewer's attention to this specific point. The sentence has been revised to clarify that women with a negative pregnancy outcome had higher levels of cortisol at an earlier egg retrieval time point. Line 422 : Also, women with a negative pregnancy outcome had higher levels of stress biomarkers such as cortisol at an earlier egg retrieval time point [23].  We believe this adjustment rectifies the confusion and accurately reflects the intended meaning.

Round 2

Reviewer 1 Report

Comments and Suggestions for Authors

To increase readers' interest in a review that would otherwise remain too obvious, a serious and complete analysis of the potential beneficial effects of some supplements should be added. This in-depth study should range from a biochemical to a clinical background, and concern not only inositol, whose effects on brain cells have been studied, but also that of other supplements. A simple paragraph is not enough to be exhaustive.

Author Response

Answers to Reviewer’s number 1 comments ( second round) : 

Thank you for your insightful comments on our manuscript entitled "[Effect of Stress on each of the Stages of the IVF Procedure: A Systematic Review]." We appreciate the constructive feedback provided, and we have carefully considered your suggestions in our revisions.

Reviewer's number 1 comment: " To increase readers' interest in a review that would otherwise remain too obvious, a serious and complete analysis of the potential beneficial effects of some supplements should be added. This in-depth study should range from a biochemical to a clinical background, and concern not only inositol, whose effects on brain cells have been studied, but also that of other supplements. A simple paragraph is not enough to be exhaustive.

"

Answer to reviewer’s number 1 first comment:

 We thank the reviewer for his suggestions and trying to improve the novelty of the paper. Please bare in mind that this review aims to analyse the effect of stress on each of the Stages of the IVF Procedure rather than the treatment of stress during IVF. We as a team did not initially intend to review the stress relieving treatments and methods of the IVF procedure and this was clearly stated in the Materials and Methods sections : “ Concerning the selection criteria, only studies including IVF cycles as an ART method were included in the research. Animal studies were excluded and articles that emphasised on minimising stress during IVF, rather than the effect of stress, were also not considered a priority.’’  Furthermore, the biochemical and clinical background of this review emphasised on the effects of chronic and acute stress during the IVf procedure. We take into account your suggestions and we have incorporated  in the discussion a thorough section of how different supplements may ameliorate stress. More specifically, in  lines  494-556 we have added all the potential beneficial effects of some supplements that may be used as treatments. Unfortunately , we were not able to find more references when it came to using supplements as a stress relieving treatment during IVF. Yet, we have incorporated  different studies that analyse these  supplements used for infertility that could possibly link these antioxidants as stress relievers. As mentioned before, all the supplements are outlined in lines 494-556  and since this input was out of the scope of this review if you have any suggestions regarding other references we would be grateful if you could assist us .

Round 3

Reviewer 1 Report

Comments and Suggestions for Authors

With the inclusion of new information on the effects of some supplements reported in the discussion, the manuscript is undoubtedly improved and ready to be published

Comments on the Quality of English Language

No comments

Author Response

We thank  the reviewers for accepting our answers to their previous comments and enhancing our manuscript (R1,R2).